# A unified Watson-Crick geometry drives transcription of six-letter expanded DNA alphabets by *E. coli* RNA polymerase

Juntaek Oh [1,2,9], Zelin Shan [3,9], Shuichi Hoshika [4,9], Jun Xu [1], Jenny Chong[1], Steven A. Benner[4] ✉, Dmitry Lyumkis [3,5,6] ✉ & Dong Wang [1,7,8] ✉

Artificially Expanded Genetic Information Systems (AEGIS) add independently replicable unnatural nucleotide pairs to the natural G:C and A:T/U pairs found in native DNA, joining the unnatural pairs through alternative modes of hydrogen bonding. Whether and how AEGIS pairs are recognized and processed by multi-subunit cellular RNA polymerases (RNAPs) remains unknown. Here, we show that *E. coli* RNAP selectively recognizes unnatural nucleobases in a six-letter expanded genetic system. High-resolution cryo-EM structures of three RNAP elongation complexes containing template-substrate UBPs reveal the shared principles behind the recognition of AEGIS and natural base pairs. In these structures, RNAPs are captured in an active state, poised to perform the chemistry step. At this point, the unnatural base pair adopts a Watson-Crick geometry, and the trigger loop is folded into an active conformation, indicating that the mechanistic principles underlying recognition and incorporation of natural base pairs also apply to AEGIS unnatural base pairs. These data validate the design philosophy of AEGIS unnatural basepairs. Further, we provide structural evidence supporting a long-standing hypothesis that pair mismatch during transcription occurs via tautomerization. Together, our work highlights the importance of Watson-Crick complementarity underlying the design principles of AEGIS base pair recognition.

All of natural life on Earth utilizes a 4-letter molecular "alphabet" to store and retrieve genetic information. In 1953, Watson and Crick revealed the structural basis of how natural bases (adenine (A), thymine (T), cytosine (C), and guanine (G)) form base pairs in DNA duplexes[1,2]. The specificity of the canonical Watson-Crick base pairs (A:T and G:C) arises from their complementary sizes and shapes, and specific hydrogen-bonding interactions[2].

The development of synthetic nucleotides that form bio-orthogonal unnatural base pairs (UBPs) for expanding the genetic alphabet has been a long-standing goal of synthetic biology[3–6]. An expanded genetic alphabet can greatly enhance the chemical diversity of natural nucleic acids, and therefore introduce new functions and properties. Several laboratories have developed diverse UBPs based on the presence (or absence) of H-bonding and shape similarity[3–6].

[1]Division of Pharmaceutical Sciences, Skaggs School of Pharmacy & Pharmaceutical Sciences, University of California, San Diego, La Jolla, CA 92093, USA. [2]Department of Pharmacy, College of Pharmacy, Kyung Hee University, Seoul 02447, Republic of Korea. [3]The Salk Institute for Biological Studies, La Jolla, CA 92037, USA. [4]Foundation for Applied Molecular Evolution, 13709 Progress Blvd Box 7, Alachua, FL 32615, USA. [5]Department of Integrative Structural and Computational Biology, The Scripps Research Institute 10550 N Torrey Pines Road, La Jolla, CA 92037, USA. [6]Graduate School of Biological Sciences, Section of Molecular Biology, University of California San Diego, La Jolla, CA 92093, USA. [7]Department of Cellular and Molecular Medicine, University of California, San Diego, La Jolla, CA 92093, USA. [8]Department of Chemistry and Biochemistry, University of California, San Diego, La Jolla, CA 92093, USA. [9]These authors contributed equally: Juntaek Oh, Zelin Shan, Shuichi Hoshika. ✉e-mail: sbenner@ffame.org; dlyumkis@salk.edu; dongwang@ucsd.edu

Early work led to the development of a series of UBPs that are joined by alternative H-bonding patterns. These are components of an Artificially Expanded Genetic Information System (AEGIS)[7–11], and currently include an expanded eight-letter alphabet with four natural and four unnatural bases (Supplementary Fig. 1 **B:S** and **P:Z**)[5]. Such alternative H-bonded UBPs can support effective DNA replication by the *Taq* DNA polymerase and RNA transcription by the single-subunit bacteriophage T7 RNA polymerase (T7 RNAP)[12,13].

However, how multi-subunit cellular RNAPs recognize AEGIS UBPs during transcription remains unknown. To employ AEGIS in cellular systems for synthetic biology applications, it is necessary to define the mechanistic principles underlying UBP recognition and transcription using multi-subunit cellular RNAPs. Therefore, the lack of mechanistic biochemical and structural insight for how AEGIS UBPs are recognized by cellular transcription machineries represents a major knowledge gap and critical barrier to further development and implementation of AEGIS UBPs for synthetic biology applications.

Here, we investigated how two components of a synthetic AEGIS base pair: **B** and **S**, can be selectively recognized and transcribed by *E. coli* RNAP in vitro. To further define the structural basis of base pair recognition during UBP transcription, we solved structures of three RNAP elongation complexes containing template–substrate UBPs in the active site. Despite the chemical and structural differences in the functional groups of the AEGIS **B:S** pair in comparison with natural base pairs, we revealed that cellular RNAP recognizes d**B:S**TP and d**S:B**TP AEGIS pairs in exactly the same manner as natural pairs. The AEGIS base pairs (d**S:B**TP and d**B:S**TP) adopt a Watson-Crick geometry within RNAP active site. Strikingly, unlike other UBPs[14,15], here we found that RNAP is captured in an active state with the ordered trigger loop in a closed conformation, which is poised for chemistry step. Furthermore, we also provide direct structural evidence to support a long-standing hypothesis that bases can adopt rare tautomeric forms that enable the formation of Watson−Crick-like (WC-like) mispairs and

therefore lead to misincorporation errors in DNA transcription. Taken together, our results yield mechanistic insights to define how UBPs are efficiently transcribed by a multi-subunit cellular RNAP and will guide further developments to reduce UBP misincorporation during transcription.

## Results

### Effective and selective recognition of B:S pair by *E. coli* RNAP during transcription

To define how AEGIS DNA is processed during bacterial transcription, we purified recombinant core *E. coli* RNAP and performed in vitro transcription assays (Fig. 1a). We observed an asymmetric transcription recognition pattern of **B:S** base pair with a strong strand bias. Specifically, we found that **B**TP is selectively incorporated opposite the d**S** in a template, while no other natural NTP incorporation was observed under identical conditions. In contrast, both **S**TP and UTP can be effectively incorporated opposite a d**B** in the template. Interestingly, for the UTP incorporation reaction, n + 2 extension product (second UTP incorporation opposite the n + 2 dA template) is also observed, suggesting efficient transcription extension from a B:U mismatched pair. These results provide important insights and guidelines for designing AEGIS-containing genome for transcription (for example, d**S** at the template strand and d**B** at the non-template strand).

To quantitively evaluate the substrate selectivity and incorporation efficiency of AEGIS UBPs by *E. coli* RNAP, we performed single-turnover kinetic assays for AEGIS UBP transcription (Fig. 1b and Supplementary Figs 2 and 3, and Table 1). Single turnover experiments allow direct determination of the principal kinetic parameters $k_{pol}$ (the pseudo-first-order catalytic rate constant) and $K_{d,app}$ (the apparent equilibrium constant for dissociation of NTPs from the RNAP elongation complex) of nucleotide incorporation. The ratio $k_{pol}/K_{d,app}$ defines the catalytic efficiency for substrate incorporation.

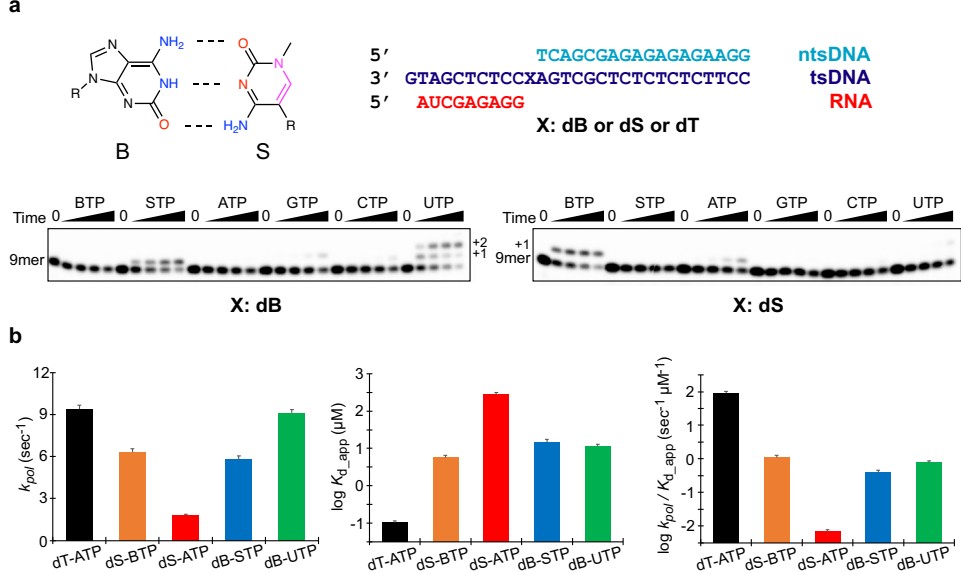

**Fig. 1 | Single nucleotide incorporation and kinetic analysis of transcription recognition of B:S pair by *E. coli* RNA polymerase. a** B:S base pair and DNA/RNA scaffold used in this study. Electron donor and accepter was colored as red and blue, respectively. Pink colored part highlights the distinct moiety from natural bases. tsDNA and ntsDNA stands for template strand DNA and non-template strand DNA, respectively. RNA, tsDNA (TS), ntsDNA (NTS) are shown in red, blue and cyan, respectively. For transcription assay, 20 μM of each nucleoside triphosphate was added, and time points were taken at 0 sec, 15 s, 1 min, 5 min and 30 min. Each reaction is repeated independently at least three times. The band of time 0 s (9mer) serves as internal molecular weight marker: p32-labeled 9mer RNA (5'-AUGGAGAGG-3'). **b** Kinetic parameters for single turnover nucleotide addition. $k_{pol}$, $K_{d,app}$ and $k_{pol}/K_{d,app}$ are shown with mean and standard error (SE) of best-fit kinetic parameters determined by nonlinear regression using Michaelis-Menten equation from time-course data points (total 81-117 data points) of different concentrations (9 − 13)(Prism 8). Kinetic data of dT-ATP, d**S**-BTP, d**S**-ATP, d**B**-**S**TP, and d**B**-UTP are shown in black, orange, red, blue, and green columns. The underlying numerical data for this graph are shown in Table 1.

We found that the $k_{pol}$ parameters of "matching" UBPs (d**S**:B TP and d**B**:S TP) are similar to those of dT:ATP. In terms of substrate binding, the $K_{d\_app}$ of d**S**:BTP, d**B**:STP and d**B**:UTP were comparable and ~50–150-fold larger than $K_{d\_app}$ of natural dT:ATP incorporation. As a result, the catalytic efficiencies ($k_{pol}/K_{d\_app}$) for d**S**:BTP and d**B**:STP are comparable, and are only two-orders of magnitude lower than that for dT:ATP incorporation.

We also measured two misincorporation scenarios using d**B**:UTP and d**S**:ATP (Fig. 1a, b). Intriguingly, the $k_{pol}$ and $K_{d\_app}$ values of d**B**:UTP misincorporation are comparable to that of matched pairs (d**S**:BTP and d**B**:STP). In sharp contrast, d**S**:ATP misincorporation has the lowest incorporation efficiency, with a major differences in both in $k_{pol}$ and $K_{d\_app}$ (Fig. 1a, b, and Table 1). Taken together, these findings reveal that the **B**:**S** pair can be effectively and selectively recognized by *E. coli* RNAP when d**S** is in the template strand.

### The B:S pair forms a Watson-Crick base pair in the active site of *E. coli* RNAP

The **B**:**S** pair shuffles the hydrogen bonding donors and acceptors that form hydrogen bonds between the components of the base pair.

### Table 1 | Kinetic parameters of natural/unnatural base incorporation

|  | Kpol (sec⁻¹) | Kd_app (μM) | Kpol / Kd_app (sec⁻¹μM⁻¹) | Relative Efficiency |
|---|---|---|---|---|
| dT: ATP | 9.4 ± 0.3 | 0.10 ± 0.01 | 92 ± 11 | 83 |
| dS: BTP | 6.3 ± 0.2 | 5.7 ± 0.9 | 1.1 ± 0.2 | 1 |
| dS: ATP | 1.8 ± 0.1 | 270 ± 40 | (6.6 ± 1.0) * 10⁻³ | 0.006 |
| dB: STP | 5.8 ± 0.3 | 15 ± 2 | 0.39 ± 0.07 | 0.35 |
| dB: UTP | 9.1 ± 0.2 | 12 ± 1 | 0.79 ± 0.09 | 0.72 |

The initial velocities were derived from time-course measurements conducted at various substrate concentrations. The kinetic parameters were obtained by plotting these initial velocities against substrate concentrations and fitting the data to the Michaelis-Menten equation.

Moreover, the **B**:**S** pair also changes the functional groups presented to the minor groove of the forming double helix. It is not clear how these changes will affect recognition during transcription.

To understand how this UBP is recognized by *E. coli* RNAP, we solved two cryo-EM structures of the *E. coli* RNAP elongation complex containing a site-specific UBP at the i + 1 site (d**S**:BTP and d**B**:STP). To obtain substrate bound structures, we utilized 3′-deoxy RNA, which allows binding of incoming nucleoside triphosphate but prevents nucleotide incorporation. The two high-resolution structures of *E. coli* RNAP with d**S**:BTP (2.70 Å) and d**B**:STP (2.65 Å) allow us to unambiguously build the UBP in the RNAP active site (Fig. 2, Supplementary Figs 4 and 5, and Supplementary Table 1).

We found that the structures of both UBPs bound in the active site of *E. coli* RNAP elongation complexes were similar to those reported previously for cognate natural substrates bound to the elongation complex (Fig. 2a)[16,17]. Despite the chemical and structural differences in the functional groups of the **B**:**S** pair (in comparison with natural base pairs), both d**S**:BTP and d**B**:STP form Watson-Crick geometry in a manner that is effectively identical to natural Watson-Crick base pairs (G:C and A:U) (Fig. 2b, c). These results indicate that the *E. coli* RNAP active site, which has evolved to recognize natural substrates, is also competent to recognize UBPs with alternative modes of hydrogen bonding.

Several DNA polymerases, such as T7 DNA polymerase (T7 DNAP), Taq DNA polymerase (Taq DNAP), and DNA polymerase beta, make direct hydrogen bond interactions with the minor groove of incoming nucleotide and upstream DNA duplex. The H-donor residues are basic or polar residues such as Lys, Arg, Gln, and Asn[18]. In sharp contrast, multi-subunit RNAPs use non-polar, hydrophobic residues as a steric gate to detect if there are any lesions at the minor groove[19,20]. Indeed, we previously reported the Pro-gate loop that interacts with incoming NTP and template base via Van der Waals interactions[19,20]. To examine the potential minor groove interactions between the **B**:**S** pair and *E. coli* RNAP, we investigated the RNAP residues near the minor groove of

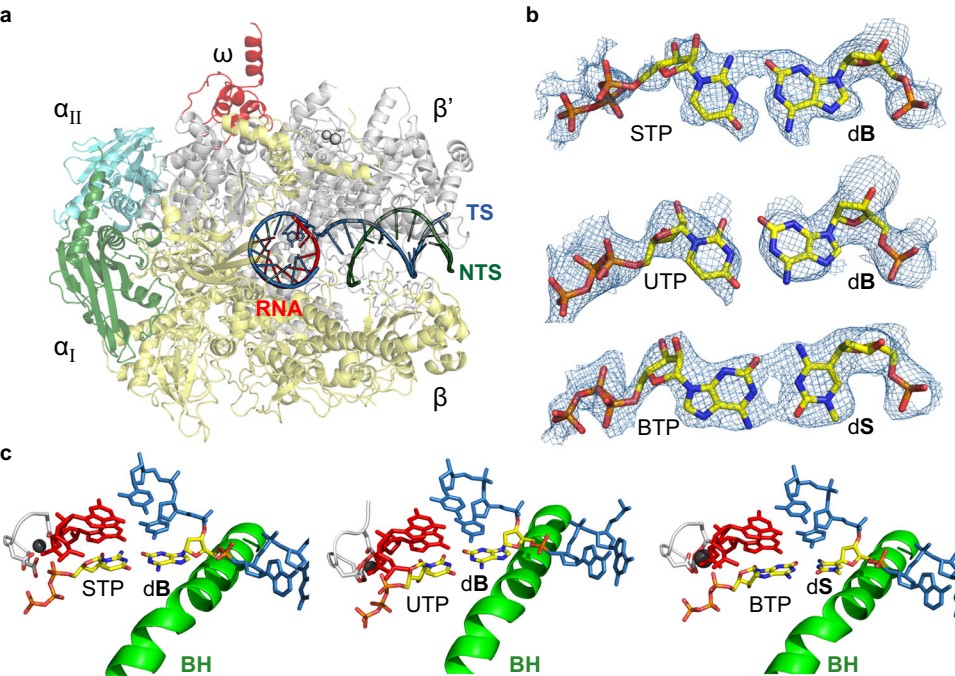

**Fig. 2 | Overall structures of *E. coli* RNA polymerase elongation complex harboring unnatural base pair. a** Overall structure of *E.coli* RNA polymerase EC. α$_I$, α$_{II}$, β′, β and ω subunit of RNAP were colored in green, cyan, gray, yellow, and red, respectively. RNA, template strand DNA (TS), non-template strand DNA (NTS) are shown in red, blue and dark green, respectively. **b** Model and map of three unnatural base pairs; d**B**:STP, d**B**:UTP and d**S**:BTP are shown with Cryo-EM density maps. **c** Active site of three UBP EC structures. Bridge helix (BH) is colored in green. UBP in +1 position were colored yellow. Other parts of the structures are colored white. The nitrogen, oxygen, phosphate, and carbon atoms of the base pair at i + 1 position are colored in blue, red, orange, and yellow, respectively.

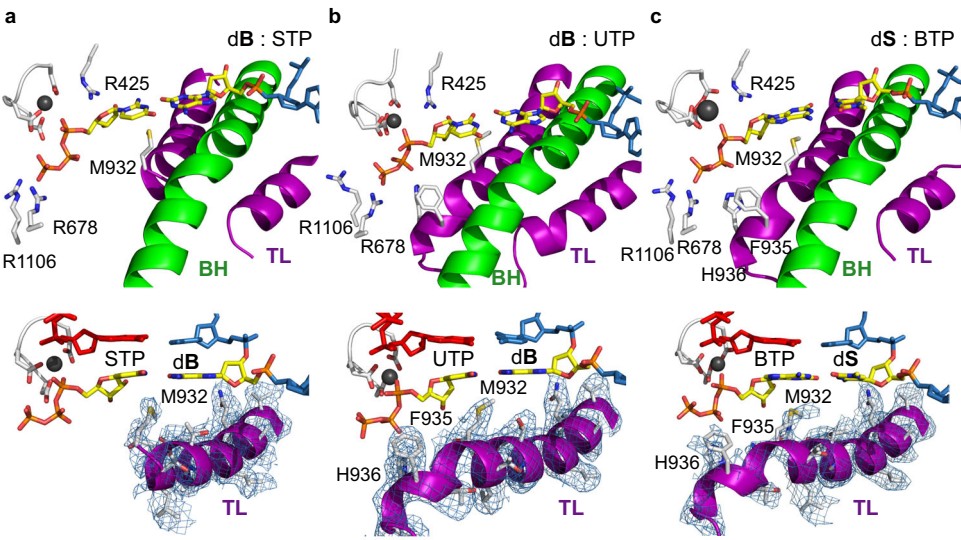

**Fig. 3 | Formation of UBP induces closure of trigger loop in *E. coli* RNAP elongation complex. a**–**c** Residues interacting with bound nucleotide triphosphate substrate are highlighted. Trigger loop (TL) and Bridge helix (BH) are colored in purple and green, respectively. Magnesium ion in black. Bottom column shows densities of TLs for (d**B**:**S**TP), (d**B**:UTP) and (d**S**:**B**TP), respectively. R678 and R1106 belongs to β subunit of RNAP. R425, M932, F935 and H936 belongs to β′ subunit of RNAP.

incoming substrate and +1 template position (Supplementary Fig. 6). We found that non-polar residues Pro427 (from Pro-gate loop) and Met1273 are close to +1 template and incoming nucleotide substrate, respectively. In addition, Ala426 (from Pro-gate loop) are close to upstream -1 base pair. These interactions are identical to natural base pair recognition by *E. coli* RNAP. These data suggest that multi-subunit RNAPs use different minor groove recognition principle than DNAPs, employing steric discrimination instead of recognition via direct hydrogen bonding interactions. Accordingly, multi-subunit RNAPs may be more tolerant to the electron donor/acceptor functional group changes at the minor groove, if they remain similar in size.

## Formation of an unnatural base pair induces trigger loop closure of *E. coli* RNAP

The natural nucleotide substrate diffuses from the secondary channel of RNAP and reaches the addition site (A-site) if it can form a Watson-Crick base pair with the template base[21,22]. Cellular RNA polymerases select the correct nucleotide substrate over a mismatched substrate via a comprehensive interaction network containing template strand, RNA primer, and protein residues at the active sites.

In particular, the trigger loop (TL), a structurally conserved motif of all multi-subunit RNAPs, is important for substrate recognition and catalysis[23–25]. TL undergoes a significant conformational change in response to nucleotide binding and addition during transcription. Binding of the correct substrate induces a conformational change to an active, closed state. Closure of TL seals off the RNAP active site and provides correct positioning of the natural nucleotide substrates for incorporation in both prokaryotic and eukaryotic RNAPs[23–25]. In contrast, TL is flexible and resides in a relaxed, inactive open state in the absence of substrate or in the presence of mismatched substrate[26].

To investigate whether the binding of unnatural nucleotides leads to a conformational change in the TL, we examined the atomic structure of RNAP active site and the TL conformation. We observed well-defined density that can be attributed to the closed TL for both d**B**:**S**TP and d**S**:**B**TP structures. We built the residues of TL corresponding to the proximal TL helix (Fig. 3). Key TL residues, such as M932, F935, H936, maintained similar interactions with incoming unnatural NTPs, compared to previously reported bacterial RNAP closed elongation complex structures containing the natural scaffold (Fig. 4 and

Supplementary Fig. 7)[17,24]. Therefore, we concluded that *E. coli* RNAP can recognize unnatural substrate **B**TP or **S**TP opposite their corresponding template partner using principles of Watson-Crick base-pairing, analogously to natural cognate NTPs. Furthermore, the high-resolution structures show that, when d**B**:**S**TP and d**S**:**B**TP are bound, the active site of RNAP is in the TL-closed, active state, which is poised for nucleotide addition. Taken together, these cryo-EM structures explain how and why STP and BTP can be effectively incorporated by *E. coli* RNAP.

## Structural basis of transcription recognition of the dB-UTP pair by *E. coli* RNAP

Our biochemical data showed that UTP can be misincorporated opposite template d**B** in an in vitro transcription assay. This efficient misincorporation prompted us to examine how UTP interacts with the d**B** template during transcription using high-resolution structural biology.

Previous reports suggest that several different forms of **B**:U base pair can arise to allow **B**:U mismatching (Supplementary Fig. 8). The first involves a **B**:U reverse wobble base pair, generated through the major keto form. The second allows for a **B**:U Watson-Crick base pair via tautomerization of either template d**B** or incoming substrate UTP as the enol tautomer[27,28]. Free energy calculations in gas and solution suggest that the d**B**:UTP (reverse) wobble pair and the d**B** (enol):UTP pair would have similar base pairing stability[27]. To distinguish between these two scenarios, we solved the cryo-EM structure of *E. coli* RNAP containing mismatched d**B**:UTP at 3.28 Å resolution and derived an atomic model from the density (Supplementary Fig. 9 and Supplementary Table 1). We found that the configuration of d**B**:UTP base pair in the active site is consistent with Watson-Crick like geometry, but not the reverse wobble base pair geometry (Fig. 2b).

Intriguingly, we also observed strong density for the TL that allowed us to fully build intact TL (β′ residues 916-1146), including the insertion motif (β′ residues 943-1130) (Fig. 4 and Supplementary Fig. 7). The map quality of TL is even better than that from matched d**B**:**S**TP and d**S**:**B**TP (Fig. 4 and Supplementary Fig. 7). Taken together, our data revealed the structural basis of effective UTP incorporation opposite the d**B** template. The d**B**:UTP base pair adopts a Watson-Crick like geometry with an active, closed TL in a manner that is analogous to a canonical matched natural base pair.

Previous free energy calculations suggest a lack of major free energy differences for distinct conformations of d**B**:T/U base pairs[27]. This raises the question: why is the Watson-Crick geometry favored for the **B**:U pair by *E. coli* RNA polymerase? To address this question, we modeled all potential base pairs within the active site (Fig. 5). We found that the d**B** (imine):UTP pair in a wobble geometry loses a hydrogen bond as well as 2′-OH interaction between ribose and R425 of β′ in pro-gate loop (Fig. 5b). On the other hand, in the reverse wobble conformation for d**B**(keto):UTP, there would be a strong steric clash between the UTP and the pro-gate loop of *E. coli* RNAP (the conserved region of yeast Pol II (β′ 425-431, rpb1 440-460 in yeast Pol II)) (Fig. 5c)[20,29]. Thus, steric constraints promote the configuration of Watson-Crick base pairing for both natural and UBPs and discourage

the reverse wobble conformation. Moreover, the Watson-Crick geometry has the strongest base stacking, with i-1 base pairs. The base stacking between -1 template DNA and d**B**, -1 RNA primer and UTP also provide further stability for Watson-Crick like pair. Taken together, these neighboring effects at RNA polymerase active site favor Watson-Crick geometry over alternative base pair conformations. Our findings reveal that Watson-Crick geometry is energetically favored during transcription. These data extend previous observations for how Watson-Crick geometry is favored during replication and translation[30–34], enhancing our understanding of the shared principles that govern the major molecular processes in the Central Dogma.

Our structure of RNAP with d**B**:UTP suggests that the d**B**:UTP base pair adopts a Watson-Crick-like geometry via tautomerization.

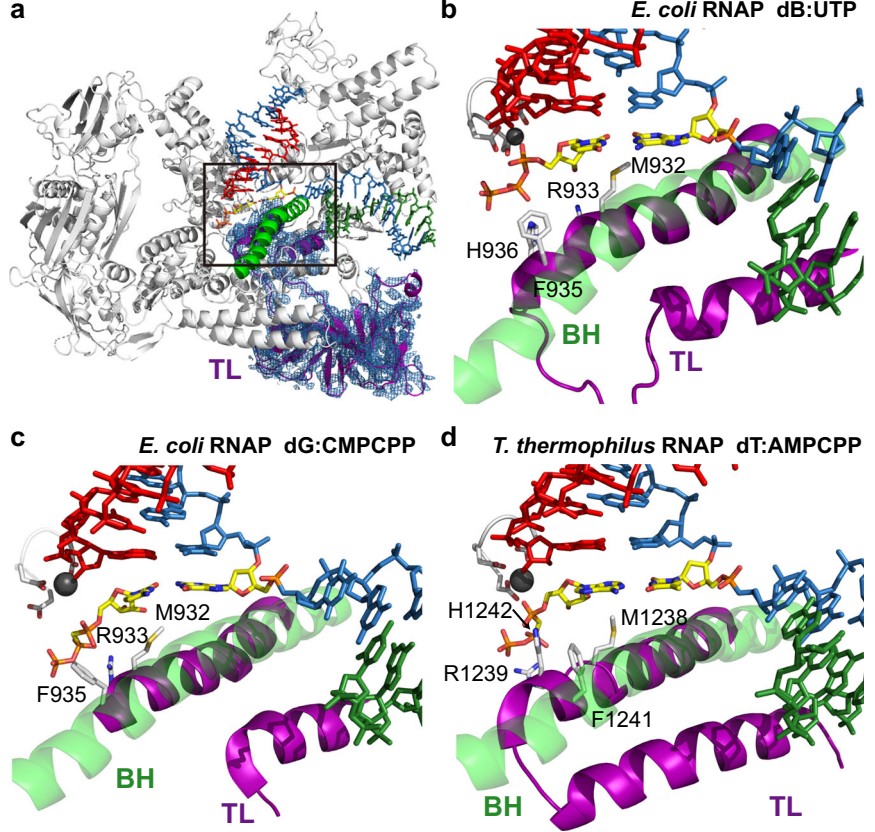

**Fig. 4 | Comparison of trigger loop configurations among different RNAPs containing unnatural and natural base pairs. a** *E. coli* RNAP with UBP. The trigger loop of *E. coli* RNAP d**B**:UTP (β′ residues 916-1146), including the insertion domain (β′ residues 943-1130) are highlighted in purple. **b–d** Comparison of three TL configurations in three RNAP EC (elongation complex) structures, including *E. coli* RNAP with d**B**:UTP (**b**), *E. coli* RNAP with dG:CMPCPP (PDB ID: 7MKO) (**c**), *T. thermophilus* RNAP with dT:AMPCPP (PDB ID: 2O5J) (**d**). All residues labeled belongs to β′ subunit of RNAP.

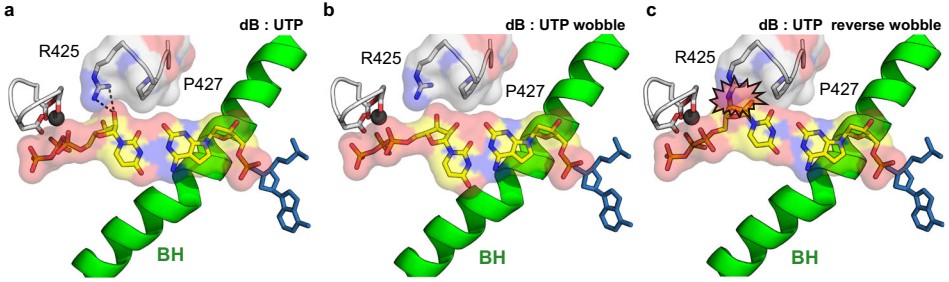

**Fig. 5 | Potential mismatched d B:UTP base pair configuration within *E. coli* RNAP active site. a** Surface representation of d**B**:UTP observed in our cryo-EM structure. **b** A d**B**:UTP wobble base pair model was prepared based on a G:U wobble base pair (PDB 6LOY). **c** A d**B**:UTP reverse wobble base pair was modeled based on a

$B_{t1}$-T reverse wobble base pair[27]. The i + 1 base pair and minor-groove edge contacting loop are highlighted in surface view. Other color code is the same as Figs. 2 and 3.

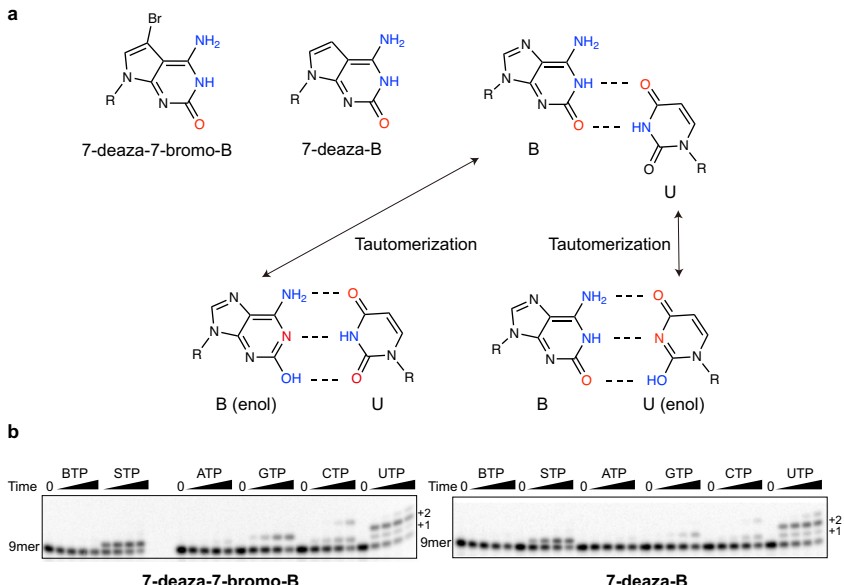

**Fig. 6 | Single nucleotide incorporation assay with 7-deaza modified dB templates. a** Schematic representation of modified **B** and possible **B**:U mismatch pairs. B (enol) tautomer:U pair and B:U (enol) tautomer were shown. Note that **B**:S and **B**:U (enol) pairs are modeled in our final structures. **b** Time-course of a single nucleotide incorporation assay: 100 μM of each nucleoside triphosphate was added and results were recorded in 0 s, 15 s, 1 min, 5 min and 30 min, respectively. Each reaction is repeated independently for three times ($n = 3$). The band of time 0 s (9mer) serves as internal molecular weight marker: p32-labeled 9mer RNA (5′-AUGGAGAGG-3′).

However, the structure, by itself, cannot distinguish which tautomerization form will prevail when there is a **B**:U mismatch pair, because both tautomers, d**B** (enol):UTP (keto) and d**B** (keto):UTP(enol), abide by principles of Watson-Crick geometry (Fig. 6a). To determine whether the keto/enol ratios of d**B** template would affect UTP incorporation, we examined the single incorporation efficiency of nucleotide triphosphates when the two keto-dominated d**B** analogs, 7-deaza-d**B** and 7-deaza-7-bromo-d**B**, reside in the template position. Previous research suggests that $K_{TAUT}$ (=[keto]/[enol]) of d**B** was ~10, meaning that ~10% of d**B** exists in the enol tautomer form, whereas two d**B** analogs 7-deaza-d**B** and 7-deaza-7-bromo-d**B** strongly favor the keto tautomer form, with the $K_{TAUT}$ values of these two d**B** analogs are $10^3$ and $10^4$, respectively (Fig. 6a)[35–38]. In our assay condition, we found that the UTP was still incorporated effectively with two keto-dominated 7-deaza-d**B** templates, suggesting the d**B** (enol) form is unlikely the major driving force for tautomerization in d**B**:UTP Watson-Crick-like base pair (Fig. 6b). Based on this biochemical evidence, we modeled d**B** (keto):UTP (enol) in our structure. Future studies would be needed to fully understand the energy landscapes and interconversion dynamic of different tautomerization forms of d**B**:UTP within the active site of RNAP.

## Discussion

### Transcription recognition of unnatural B:S base pair as a third base pair

Our biochemical and structural analyses demonstrate that the **B:S** pair can be utilized effectively and selectively by *E. coli* RNAP as a third base pair in a manner similar to that used by natural base pairs during transcription. We show that **B**TP and **S**TP can be incorporated efficiently opposite the template d**S** and d**B**, respectively.

Strikingly, our structural studies reveal that the trigger loop can adopt a fully closed active state in all cryo-EM structures. The configuration of TL is fine-tuned to discern both matched and mismatched substrates, allowing it to play an important role in nucleotide selection, positioning, and addition. These structures suggest that *E. coli* RNAP can recognize the **B:S** UBPs in the same manner as natural base pairs, highlighting the potential of **B:S** pair as a third base pair for an

expanded genetic alphabet. Our data yield mechanistic insights into how AEGIS UBPs with alternative hydrogen bonding are specifically recognized and incorporated into an elongating RNA transcript by a multi-subunit cellular RNA polymerase. Importantly, these AEGIS UBPs can be recognized just like natural base pairs, and we are able to capture the active, closed state of RNAP elongation complex. These insights will guide the design of next generation AEGIS UBPs and facilitate engineering of UBP-specific RNA polymerases. Thus, our results expand the area of synthetic biology, which will ultimately help design de novo proteins using non-canonical amino acids.

In addition to UBP with alternative hydrogen bonding developed by Benner group, other groups have developed hydrophobic UBPs without any inter-base hydrogen bonding. These include the **TPT3:NaM** pair by Romesberg group and the **Ds:Pa** pair by Hirao group[14,15]. Recent studies showed that the hydrophobic **TPT3:NaM** pair can be also transcribed by *E. coli* RNAP to produce mRNA with unnatural codons, which can be translated in proteins with site-specifically incorporated noncanonical amino acids[39]. However, the rate of incorporation of these hydrophobic UBPs are much slower than that of the **S:B** pair. Our understanding for UBP transcription is still very limited. Recently, our group reported the structural basis of transcription recognition of hydrophobic UBPs (**TPT3:NaM** pair by RNA Pol II and **Ds:Pa** pair by T7 RNAP)[14,15], none of these previous RNAP structures were captured in a closed, active state. No *E. coli* RNAP structures containing hydrophobic UBPs are yet available for a direct comparison with AEGIS UBPs. Systematic biochemical and structural studies are needed to compare transcription recognition of different UBPs by the same enzyme, or transcription recognition of the same UBPs by different RNA polymerases for future studies.

### Structural evidence of mismatch base pairs via tautomerization during transcription

Mismatches of natural base pairs can arise and lead to mutations during replication, transcription, and translation. Watson and Crick proposed that spontaneous mutations may stochastically arise when an incoming base adopts an unfavorable tautomeric form[2]. The

environment surrounding the base pair undoubtedly impacts the tautomeric equilibrium between keto/enol forms. In the last decade, WC-like mispairs via tautomerization have been observed in the active sites of DNA polymerases and the ribosomes in catalytically competent conformations[30–32,34,40–43]. These results provide important insights into misincorporations at replication and translation processes and support Watson and Crick's tautomer hypothesis.

However, it remained unclear whether the similar tautomerization mechanism for mismatched pairs may also arise during transcription, due to the lack of structural data. Our results demonstrate that the same tautomerization principles can be extended into the transcription process. Here, we provide evidence that UTP can tautomerize, leading to a base pair structure formed by three hydrogen bonds. Our observation that the **B**:U base pair adopts the Watson-Crick U's enol form, and not the wobble base pair U's keto form, serves as an example of base pair tautomerization during transcription. Therefore, our results provide structural evidence to support the long-standing tautomer hypothesis.

## Methods

### Oligonucleotides and nucleotides used in this study

RNA oligo (5′-AUCGAGAGG), 3′-deoxy RNA oligo (5′-AUCGAGAG/3′dG/), template strand (ts) DNA oligo (dT, 5′- CCTTCTCTCTCTCGCTGAT CCTCTCGATG) and non-template strand (nts) DNA oligo (5′-TCAGCGA GAGAGAGAAGG) were purchased from IDT. AEGIS ribotriphosphates (r**B**TP and r**S**TP) were from Firebird Biomolecular Sciences LLC (Alachua, FL). Standard phosphoramidites (Bz-dA, Ac-dC, dmf-dG, dT and d**B**: dmf-isodG-CE Phosphoramidite) and controlled pore glass (CPG) having standard residues were purchased from Glen Research (Sterling, VA) and AEGIS phosphoramidites (d**S**, 7-deaza-d**B** and 7-bromo-7-deaza-d**B**) were purchased from Firebird Biomolecular Sciences LLC (Alachua, FL).

All oligonucleotides containing d**S**, d**B**, 7-deaza-d**B**, and 7-bromo-7-deaza-d**B** were synthesized on an ABI 394 DNA Synthesizer following standard phosphoramidite chemistry[44,45]. Briefly, samples of CPG supporting the synthetic oligonucleotides were treated with 2.0 mL of 1 M DBU in anhydrous acetonitrile at room temperature for 24 h to remove the NPE group. Then, the CPGs were filtered, dried, and treated with concentrated ammonium hydroxide at 55 °C for 16 h. After removal of ammonium hydroxide, the oligonucleotides were purified on an ion-exchange HPLC, and then desalted using Sep-Pac® Plus C18 cartridges (Waters).

### Purification of *E. coli* RNA polymerase

The expression plasmids for *E. coli* RNA polymerase (deposited by Seth Darst lab) were purchased from Addgene. We followed previous purification methods with modification[46]. pEcRNAP6 (Addgene # 128940) and pACYCDuet-LEc-rpoZ (Addgene # 128837) were co-transformed to BL21 (DE3) competent cells (Novagen). 0.25 mM of IPTG was added to cell culture when O.D. reached 0.6 and protein were expressed in 20 °C for overnight. Cells were collected and resuspended in Buffer A [50 mM Tris (pH 7.4), 500 mM NaCl, 5% (v/v) glycerol, 2 mM β-mercaptoethanol (BME) and 1X EDTA-free protease inhibitor (Sigma-Aldrich)]. After lysis and centrifugation, cell lysate was loaded to Ni-NTA resin and washed with Buffer A + 20 mM imidazole, Buffer A (300 mM NaCl) + 30 mM imidazole and eluted by Buffer A (300 mM NaCl) + 250 mM imidazole. Eluted sample was diluted with Buffer A (no salt) 2 times to reduce NaCl concentration to 150 mM. After dilution, sample was loaded to HiTrap Heparin column, and purified by increasing NaCl concentration from 150 mM to 800 mM. For anion exchange purification, heparin elute was dialyzed against Buffer B (20 mM HEPES pH 8.0, 100 mM NaCl, 5% (v/v) glycerol, 1 mM DTT). After dialysis, sample was loaded into HiTrap Q column. *E. coli* RNA polymerase (α₂β′βω) was purified by increasing NaCl concentration from 100 mM to 800 mM. During concentration, buffer was changed

to Buffer B with 150 mM NaCl, flash frozen in liquid nitrogen and stored in −80 °C for further use.

### In vitro transcription assay and kinetic analysis

Transcription assays were performed with previously described methods with minor modifications[19]. Mini-scaffold containing 200 nM RNA (P³²-labeled), 400 nM tsDNA and 600 nM ntsDNA were mixed in elongation buffer (EB), 20 mM Tris (pH 8.0 at 4 °C), 40 mM KCl, 5 mM DTT and 5 mM MgCl₂. Scaffold was annealed by heating at 80 °C for 5 min and slow cooled down to room temperature for at least 2 h. *E. coli* RNAP elongation complex was prepared by adding Mini-scaffold in 1X EB and incubate for 20 min in 30 °C. Reaction was initiated by adding elongation complex to each nucleotide triphosphate. As a result, reaction sample has 20 nM of Mini-scaffold, 120 nM *E. coli* RNAP and 20 μM each nucleotide triphosphates in 1X EB, if not mentioned otherwise. At each timepoint, the reaction mixture was pipetted into stop buffer [90% (v/v) formamide, 50 mM EDTA, 0.05% (w/v) xylene cyanol and 0.05% (w/v) bromophenol blue]. All samples were denatured by heating 95 °C for 10 min and analyzed by 12% denaturing urea-PAGE gel analysis. All reactions were performed in room temperature. Kinetic parameters in Table 1 and Supplementary Fig. 2 were obtained by using Prism8 non-linear regression fit (Michaelis-Menten). Briefly, non-linear Michaelis-Menten regression was performed by plotting [S] vs V0 values, which obtained from time vs % incorporated substrate graph in Supplementary Fig. 2 (regression curves) and 3 (Raw gel data). Each concentration has 9 time points. Time-course data points used for kinetic parameters fitting are: total 81 data points from 9 different concentrations for dT:ATP, total 117 data points from 13 different concentrations for d**S**:**B**TP, total 90 data points from 10 different concentrations for d**S**:ATP, total 99 data points from 11 different concentrations for d**B**:**S**TP and total 81 data points from 9 different concentrations for d**B**:UTP, respectively.

### Preparation of *E. coli* RNA polymerase elongation complex for electron microscopy

Mini-scaffold containing 3′-deoxy RNA, tsDNA and ntsDNA with molar ratio of 1.2:1:1.2 were annealed in 1X EB. To form elongation complex, purified *E. coli* RNAP were mixed with prepared Mini-scaffold with molar ratio 1:1.3 and incubated in ice for 1 h (pH 8.0 at 4 °C). Final 4 mM of MgCl₂ and **B**TP or **S**TP or UTP in 1X EB were added to elongation complex and incubated in ice for 30 min. Before sample preparation, final 4 mM CHAPSO dissolved in 1X EB were added to complex to reduce particle orientation bias[47]. The final *E. coli* RNAP concentration was 15–18 mg/ml. If dilution was needed, 4 mM substrate nucleotide triphosphate, 4 mM MgCl₂, and 4 mM CHAPSO in 1X EB were used.

**Cryo-EM sample preparation.** The procedures for grid vitrification for all three cryo-EM sample preparations were identical. 2.5 μl of 15–18 mg/ml *E. coli* RNA polymerase elongation complex was pipetted onto freshly plasma cleaned (20 s, Solarus plasma cleaner) holey grids (Quantifoil UltrAuFoil R 1.2/1.3 300 mesh) at 4 °C in the cold room. The sample was incubated onto grids for 30 s before manually blotting for 6 s, then plunged into the liquid ethane using a manual plunger. The vitrified grids were transferred to liquid nitrogen for storage and data collection.

**Cryo-EM data collection and processing d S.** **B**TP and d**B**:**S**TP cryo-EM datasets were collected in Stanford-SLAC cryo-EM center, while d**B**:UTP cryo-EM dataset was collected in the Scripps cryo-EM facility. All datasets were collected using a Titan Krios transmission electron microscope (Thermo Fisher Scientific) operating at 300 keV, equipped with either a K2 or K3 direct electron detector (Gatan) with a GIF Quantum Filter with a slit width of 20 eV (an energy filter was not used for d**B**:UTP sample). All data collections were performed either using SerialEM, EPU or Leginon[48–51]. The software used for data collection

depended on what was installed for the microscope. Some data collections were performed using stage tilt[52]. The imaging parameters for each dataset are comprehensively summarized in Supplementary Table 1.

All three cryo-EM datasets were processed using a similar workflow. For each dataset, the movies were imported into Relion 4.0-beta-2 for dose-weighted motion correction on 5-by-5 patch squares and with a B factor of 150 Å² using the gain reference that was generated during data collection[53,54]. The motion-corrected micrographs were then imported into Warp 1.0.9 to perform CTF estimation and particle selection[55]. The particles that scored above 0.7 were selected in Warp with a re-trained BoxNet model using constraint settings of particle diameter of 180 Å and a minimum distance between particles of 20 Å. The particle star file was made by initially extracting selected particles using a box size of 384 pixels.

The aligned micrographs in Relion and particle star file from Warp were imported into cryoSPARC V3.3.2 for the next steps of data processing[56]. The particles were re-extracted in cryoSPARC with the same box size of 384 pixels for performing 2D classification. The particles that contributed to the best classes with good particle features were selected after several rounds of 2D classification. The selected particles from 2D were subjected to multiple rounds of heterogeneous refinement to further clean the dataset until no further improvements were observed, as indicated by the Fourier shell correlation (FSC)[57].

After each round of heterogeneous refinement, particles that were assigned to the highest-resolution map were selected, whereas the remaining particles were discarded. Multiple rounds of heterogeneous refinement were performed. The selected particles from the last heterogeneous refinement were used to perform a homogeneous refinement to generate a high-resolution reconstruction. At this point, the remaining particles from homogeneous refinement were imported back to Relion for Bayesian polishing using default parameters[58]. Subsequently, the polished particles were re-imported back to cryoSPARC to perform one round of per-particle CTF refinement and one round of global CTF refinement. We repeated the procedure – Bayesian polishing in Relion followed by CTF refinement in cryoSPARC iteratively, until no further improvements in resolution were observed, as determined using the FSC. Generally, the optics parameters, per-particle defoci, and the map resolutions converged after several rounds of polishing and CTF refinement.

The final map was generated by the last homogeneous refinement step. Density modification was performed for high-resolution d**S:B**TP and d**B:S**TP maps in Phenix-1.20.-4487 prior to modeling[59,60]. The directional resolution of the map was evaluated using the 3D FSC server (3dfsc.salk.edu)[52], and the quality of the orientation distribution was evaluated using the Sampling Compensation Function (SCF)[61,62]. All images in Supplementary Figs 4, 5, and 9 were generated using UCSF Chimera[63].

## Model building

We first built the atomic model for d**B:S**TP map, which was the highest-resolution map obtained during our experiments. The initial unhydrated atomic model of *E.coli* RNA polymerase elongation complex, resolved to 4.4 Å (PDB 6ALH was rigid body docked into the density modified d**B:S**TP map using UCSF Chimera[16]. Clear density in the active site of *E.coli* RNA polymerase allowed unambiguous assignment of register of the DNA template and RNA oligomer that used for assembling d**B:S**TP RNA polymerase elongation complex.

For unnatural nucleotide r**S**TP or r**B**TP, the coordinates and restraint files were generated by using phenix elbow, then atomic model of r**S**TP was fitted to the map[64]. The docked atomic model of RNA polymerase elongation complex was merged with the docked atomic model of r**S**TP to obtain the first intact rigid body docking model for d**B:S**TP map. Subsequently, the model was manually adjusted in Coot wherever discrepancies were apparent in the

high-resolution map[65]. The final model was generated by iterative model adjustment using Coot and real-space refinement using phenix with restraint weights[64]. For modeling d**S:B**TP and d**B:**UTP maps, the same procedure was adopted, except using d**B:S**TP atomic model as the starting model.

d**S:B**TP and d**B:S**TP atomic models were individually hydrated to account for their high-resolution using phenix.douse. Before dousing, cryo-EM density maps were resampled and resized such that the voxel size became roughly one quarter of the resolution estimate using software available in CisTEM[66]. We next used phenix.douse in Phenix 1.20.1-4487 to programmatically add water molecules using settings (dist_min = 2.5 dist_max = 4.5 keep_input_water = TRUE sphericity_filter = false, varied in map_threshold value) to the density that did not account for protein and DNA molecules. The hydrated atomic models were individually subjected to real space refinement using phenix.real_space_refinement.

After real space refinement, the hydrated atomic models were individually inspected in Coot to evaluate agreement of each water placement within cryo-EM maps, and problematic water molecules were removed if they (I) were placed in vacant density, (II) clashed with protein and nucleotide molecules, (III) were placed in densities that belong to protein and nucleotide molecules, or (IV) were in periphery region that is far away from intact complex. We performed several iterative rounds of real space refinement in Phenix and water evaluation in Coot. The geometry of the final models and validation statistics were generated by Molprobity[67]. The refinement statistics for each structural model are summarized in Supplementary Table 1. All the structural figures were prepared using PyMOL[68].

## Reporting summary

Further information on research design is available in the Nature Portfolio Reporting Summary linked to this article.

## Data availability

The electron potential maps of RNAP elongation complexes are deposited into the electron microscopy databank as EMD-40862, EMD-40863, and EMD-40864. The models for RNAP elongation complexes have been deposited into the PDB as PDB ID 8SY5 (CryoEM structure of *E.coli* RNA polymerase elongation complex containing d**S** template and **B**TP as substrate), PDB ID 8SY6 (CryoEM structure of *E.coli* RNA polymerase elongation complex containing d**B** template and UTP as substrate), and PDB ID 8SY7 (CryoEM structure of *E.coli* RNA polymerase elongation complex containing d**B** template and **S**TP as substrate). Other cited published PDB entries are: PDB entries 6ALH (CryoEM structure of *E.coli* RNA polymerase elongation complex), 7MKO (*E.coli* RNA polymerase elongation complex), 2O5J (Crystal structure of the *T. thermophilus* RNAP polymerase elongation complex with the NTP substrate analog), 6L0Y (Structure of dsRNA with G-U wobble base pairs). Source data are provided with this paper.

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

## Acknowledgements

This work was supported by grants from the National Institutes of Health (R01 GM102362 and R01 GM148476 to D.W., R21CA251043 to S.H., P30 CA01495 and U54 AI170855 to D.L.), from the National Science Foundation (MCB-2048095 to D.L. and MCB-2123995 to S.A.B.). D.L. also acknowledges support from the Hearst Foundations and the Margaret T. Morris Foundation. The cryo-EM data were collected at the cryo-EM S2C2 facility of SLAC National Accelerator Laboratory, which is supported by S2C2 grant U24 GM129541 and at Scripps Research (supported by S10 OD032467).

## Author contributions

S.A.B., D.L. and D.W. designed research; J.O., Z.S., S.H., J.X. and J.C. performed research; J.O., Z.S., S.H., S.A.B., D.L. and D.W. analyzed data; and J.O., Z.S., S.H., S.A.B., D.L. and D.W. wrote the paper.

## Competing interests

The authors declare no competing interests.
