## [Peer Review File · Nature Communications]

Reviewers' Comments:

Reviewer #1:

Remarks to the Author:

In this manuscript, the authors investigate if and how the E. coli RNA polymerase recognizes the S-B unnatural base pair developed by the Benner group. They first report kinetic assays that demonstrate recognition. The efficiency of recognition under the employed conditions is somewhat less than a natural base pair (this is certainly also true for the other unnatural base pairs that have been developed) and that specificity is limited by the efficient incorporation of U opposite dB in the template. In general, the RNAP recognizes S-B very much in the same way as it recognizes a natural base pair, including specific interactions with the nucleotides and closure of the trigger loop (which is unique among unnatural base pairs). These studies represent further elegant support of the underlying rationale for the design of these unnatural base pairs – the mimicry of a natural base pair, but with an orthogonal hydrogen bonding topology.

The authors also report the very interesting result that tautomerization, forced by steric constraints in the active site, underlies the relatively efficient insertion of UTP opposite dB. The role of tautomerization in replication, transcription, and translation has long been debated, and this result will be of significant interest to a broad range of researchers.

This manuscript should be accepted for publication, and I offer the following comments for the authors to consider.

1) In the first sentence of the introduction, "life" should be replaced with "natural life," and the authors should be careful regarding the use of terms like "cellular systems" as the work is all in vitro.

2) The authors use the term "evolutionary advanced," to describe the E. coli RNAP. It is unclear to me what this means as everything in nature has been evolving for the same amount of time. Perhaps another description might be selected to make the authors' point, such as its being a multi-subunit or more general RNAP.

3) At several places the authors state or imply that the "chemistries" and structures of the unnatural pair are very distinct from a natural pair. I feel that this obfuscates the very design principle used for their development. I would prefer to see the story told as a confirmation of the design principle.

4) $k_{pol}/K_d,app$ does not define the substrate specificity, it is of course the ratios of $k_{pol}/K_d,app$ for correct and incorrect synthesis that defines specificity.

5) While the interactions are shown in the Figures, I think a little more discussion of the specific interactions between the nucleotides and the polymerase (especially in the developing minor groove) would be interesting and helpful to the reader.

6) The authors make an argument that tautomerization is more likely to be occurring in the incoming triphosphates as opposed to the nucleobase in the template. Can they provide any discussion of the interactions mediated by the polymerase that might favor this. Do the authors think that this might be true in general for natural nucleotides?

7) I feel that the statement that the authors have completed our understanding of how shared principles govern the major processes in the central dogma, should be dialed back a little.

Reviewer #2:

Remarks to the Author:

In this manuscript, the authors use biochemical analyses and cryo-electron microscopy structures to elucidate the mechanistic principles for unnatural base pair recognition (and incorporation) by E.

coli RNA polymerase (RNAP). The authors also convincingly show that mismatch base pairs can be incorporated by E. coli RNAP due to tautomerization of the natural UTP substrate. This work is very interesting, very well presented, and the biochemical and structural work are of the highest quality. I only have one issue that needs to be addressed by the authors prior to publication:

- In the cryo-EM structures, elongation complexes of E. coli RNAP were assembled in the presence of triphosphate substrates (either STP, UTP, or BTP, as shown in Fig. 2). In the structures, the authors trapped a form of the RNAP (with the Trigger-Loop closed, or folded) that is poised for catalysis, but according to the authors structural modeling and from the cryo-EM maps presented, it does not appear that catalysis has proceeded. What is inhibiting catalysis in these samples? At least according to the Materials and Methods, a 3'-deoxy RNA primer oligo was not used.

Reviewer #3:

Remarks to the Author:

The manuscript "A Unified Watson-Crick Geometry Rule: Structural Basis of Transcription Recognition of a Six-Letter Expanded Genetic Alphabet by Cellular RNA Polymerase by Wang et al. provides a kinetic and structural perspective on the recognition of unnatural base pairs (UBPs, specifically S (isoG) and B (methylisoC) by the multi-subunit cellular transcription machinery polymerase from E. coli RNA. The work elucidates the transcription of the SB UBP by determining kinetic parameters for incorporation of S or the natural nucleotides opposite dB in the template and for incorporation of B or the natural nucleotides opposite dS in the template. The authors find that indeed dS directs the incorporation B with better efficiency than any of the natural nucleotides, and dB directs the incorporation of both S and U with similar efficiency. CryoEM structures of the corresponding ternary complexes indicate that the corresponding active sites are in a closed, active state conformation. Additionally, the structures strongly suggest that the incorporation of U opposite dB involves tautomer formation rather than wobble pair formation. Overall, this work provides key new insights about how polymerases respond to unnatural base pairs and given the growing importance of UBPs for synthetic biology and nucleic acid biochemistry, these types of detailed structural and kinetic studies are important for revealing the principles by which natural polymerase work and for generating leads regarding how to improve them. I believe this work will be of broad interest to the Nature readership. Some specific comments follow:

The authors note on page 8 that their cryoEM structures "explain how and why STP and BTP can be effectively incorporated by E. coli RNAP." They also note that the UBP incorporation occurs two orders of magnitude slower than AT/TA, resulting from primarily a difference in K_d of the NTP for the ternary complex. Can the authors provide any explanation for the weaker binding from their structures? Although beyond the scope of this paper, it seems that it will be important in the future to understand the biophysical origin of this weaker binding, as it could hold the clue to developing higher fidelity/efficiency UBP/RNAP systems.

Text should make clear that the PZ pair, AT gels, and B(imine tautomer) are present in Supporting information.

Page 9 –provide information in the text about pH and temperature of the complex before freezing as these can affect tautomerization

- Fig 3. Should be Fig. 4 in the text (2 times)

Page 15 – "cooled-down" instead of "cool-down"?

Page 16 – they should include the percentage of dPAGE gel

- "Watson-Crick U's enol form", "wobble base pair U's keto form"

Figure 1: Explain abbreviations in the legend – nts, ts

X: dB or dS (what about dT they checked with ATP?)

Figure 6: Add information to clarify what tautomerizes – "B tautomerization" "U tautomerization" instead of just "tautomerization"

UTP is incorporated in a different manner than other nucleotides (several bands visible) – needs explanation

The table showing kinetic parameters should have an explanatory title/legend/ and footnotes.

REVIEWER COMMENTS

Reviewer #1 (Remarks to the Author):

In this manuscript, the authors investigate if and how the *E. coli* RNA polymerase recognizes the S-B unnatural base pair developed by the Benner group. They first report kinetic assays that demonstrate recognition. The efficiency of recognition under the employed conditions is somewhat less than a natural base pair (this is certainly also true for the other unnatural base pairs that have been developed) and that specificity is limited by the efficient incorporation of U opposite dB in the template. In general, the RNAP recognizes S-B very much in the same way as it recognizes a natural base pair, including specific interactions with the nucleotides and closure of the trigger loop (which is unique among unnatural base pairs). These studies represent further elegant support of the underlying rationale for the design of these unnatural base pairs – the mimicry of a natural base pair, but with an orthogonal hydrogen bonding topology.

The authors also report the very interesting result that tautomerization, forced by steric constraints in the active site, underlies the relatively efficient insertion of UTP opposite dB. The role of tautomerization in replication, transcription, and translation has long been debated, and this result will be of significant interest to a broad range of researchers.

This manuscript should be accepted for publication, and I offer the following comments for the authors to consider.

We appreciate the positive comments on our manuscript.

1) In the first sentence of the introduction, “life” should be replaced with “natural life,” and the authors should be careful regarding the use of terms like “cellular systems” as the work is all *in vitro*.

As suggested by the reviewer, we replaced “life” to “natural life”, and mentioned we utilized *in vitro* system for our biochemical assays.

2) The authors use the term “evolutionary advanced,” to describe the *E. coli* RNAP. It is unclear to me what this means as everything in nature has been evolving for the same amount of time. Perhaps another description might be selected to make the authors' point, such as its being a multi-subunit or more general RNAP.

We agree to Reviewer's comment that evolution is not a concept that can distinguish superiority and inferiority. Therefore, we followed the reviewer's suggestion and changed the term “evolutionary advanced” to “multi-subunit RNAP”.

3) At several places the authors state or imply that the “chemistries” and structures of the unnatural pair are very distinct from a natural pair. I feel that this obfuscates the very design principle used for their development. I would prefer to see the story told as a confirmation of the design principle.

We agree with the reviewer, and we revised our abstract to emphasize that our data validate the design principle of AEGIS UBP.

4) $k_{pol}/K_{d,app}$ does not define the substrate specificity, it is of course the ratios of $k_{pol}/K_{d,app}$ for correct and incorrect synthesis that defines specificity.

We revised that sentence to mention that $k_{pol}/K_{d,app}$ defines catalytic efficiency for substrate incorporation, and use ratio of $k_{pol}/K_{d,app}$ values of correct and incorrect incorporation to discuss about substrate specificity.

5) While the interactions are shown in the Figures, I think a little more discussion of the specific interactions between the nucleotides and the polymerase (especially in the developing minor groove) would be interesting and helpful to the reader.

As suggested by the reviewer, we created a new Extended Data Figure 6 to highlight polymerase active site residues (such as Pro in the pro-gate region) that are close to the minor groove of the RNA:DNA hybrid and nucleotides. We also included a paragraph to discuss about the potential interactions between polymerases and the minor groove of RNA primer and nucleotide substrate (also see below).

6) The authors make an argument that tautomerization is more likely to be occurring in the incoming triphosphates as opposed to the nucleobase in the template. Can they provide any discussion of the interactions mediated by the polymerase that might favor this. Do the authors think that this might be true in general for natural nucleotides?

To figure out whether change of “B tautomerization” would affect UTP incorporation, we used 7-deaza-dB (that is, 7-deazaisoguanosine) and 7-deaza-7-bromo-dB, to replace isoguanosine (dB). These 7deaza-dB analogs exhibit intrinsically much less tautomerization than the dB template (Page 11 and Figure 6a). Nevertheless, even with lower tautomerization of these UBP templates, we observed efficient UTP incorporation, suggesting that UTP can still tautomerize to fit into Watson-Crick base pair geometry. However, we didn’t intend to generalize our statement that tautomerization is more likely to be occurring in the incoming triphosphates as opposed to the nucleobase in the template. We also don’t know whether incoming substrate always tautomerizes during mismatched incorporation of natural base pairs. To address this question, further experiments using natural base pair are required, which is our future project. Currently, we

hypothesize that both the template and incoming substrate may tautomerize if the tautomer is able to form Watson-Crick base pairs (which fits better within the polymerase active site).

In term of potential interactions, we found that Pro 427 from Pro-gate loop and Met 1273 that are close to the minor groove edge of +1 template nucleobase and incoming NTP and R425 interact with the sugar moiety of UTP (Figure 5C). These interactions may favor UTP tautomerization to fit Watson-Crick base pair geometry instead of alternative reversible wobble base pair geometry (See new Extended Data Figure 6 and Figure 5C).

7) I feel that the statement that the authors have completed our understanding of how shared principles govern the major processes in the central dogma, should be dialed back a little.

We followed the reviewer's suggestion and toned down our statement to "enhancing our understanding of the shared principles that govern the major molecular processes in the Central Dogma".

Reviewer #2 (Remarks to the Author):

In this manuscript, the authors use biochemical analyses and cryo-electron microscopy structures to elucidate the mechanistic principles for unnatural base pair recognition (and incorporation) by E. coli RNA polymerase (RNAP). The authors also convincingly show that mismatch base pairs can be incorporated by E. coli RNAP due to tautomerization of the natural UTP substrate. This work is very interesting, very well presented, and the biochemical and structural work are of the highest quality. I only have one issue that needs to be addressed by the authors prior to publication:

We appreciate for the positive comments on our manuscript. We added sequence information that were used in our cryo-EM structures.

- In the cryo-EM structures, elongation complexes of E. coli RNAP were assembled in the presence of triphosphate substrates (either STP, UTP, or BTP, as shown in Fig. 2). In the structures, the authors trapped a form of the RNAP (with the Trigger-Loop closed, or folded) that is poised for catalysis, but according to the authors structural modeling and from the cryo-EM maps presented, it does not appear that catalysis has proceeded. What is inhibiting catalysis in these samples? At least according to the Materials and Methods, a 3'-deoxy RNA primer oligo was not used.

We apologize for this confusion. Indeed, we utilized 3'-deoxy RNA primer (5'-AUCGAGAG/3'dG/) to prevent nucleotide addition and obtain substrate bound structure. Our deposited PDB already had 3'-deoxy RNA information. However, we did not indicate this in our materials and methods section. We added the sequence information of the 3'-deoxy RNA in

methods section and also added a sentence that we utilized 3'-deoxy RNA for structural studies in result section.

Reviewer #3 (Remarks to the Author):

The manuscript “A Unified Watson-Crick Geometry Rule: Structural Basis of Transcription Recognition of a Six-Letter Expanded Genetic Alphabet by Cellular RNA Polymerase by Wang et al. provides a kinetic and structural perspective on the recognition of unnatural base pairs (UBPs, specifically S (isoG) and B (methylisoC) by the multi-subunit cellular transcription machinery polymerase from *E. coli* RNA. The work elucidates the transcription of the SB UBP by determining kinetic parameters for incorporation of S or the natural nucleotides opposite dB in the template and for incorporation of B or the natural nucleotides opposite dS in the template. The authors find that indeed dS directs the incorporation B with better efficiency than any of the natural nucleotides, and dB directs the incorporation of both S and U with similar efficiency. CryoEM structures of the corresponding ternary complexes indicate that the corresponding active sites are in a closed, active state conformation. Additionally, the structures strongly suggest that the incorporation of U opposite dB involves tautomer formation rather than wobble pair formation. Overall, this work provides key new insights about how polymerases respond to unnatural base pairs and given the growing importance of UBPs for synthetic biology and nucleic acid biochemistry, these types of detailed structural and kinetic studies are important for revealing the principles by which natural polymerase work and for generating leads regarding how to improve them. I believe this work will be of broad interest to the Nature readership. Some specific comments follow:

We appreciate for the positive comments on our manuscript. We revised our manuscripts as suggested by reviewer. Detailed responses are provided below:

The authors note on page 8 that their cryoEM structures “explain how and why STP and BTP can be effectively incorporated by *E. coli* RNAP. “ They also note that the UBP incorporation occurs two orders of magnitude slower than AT/TA, resulting from primarily a difference in K_d of the NTP for the ternary complex. Can the authors provide any explanation for the weaker binding from their structures? Although beyond the scope of this paper, it seems that it will be important in the future to understand the biophysical origin of this weaker binding, as it could hold the clue to developing higher fidelity/efficiency UBP/RNAP systems.

As the reviewer indicated, we also noticed this interesting result: K_d values for unnatural substrate are indeed somewhat higher than that of natural substrates. Currently, we do not fully understand where this weak binding comes from. It seems reasonable to assume that natural RNAP has evolved to specifically recognize and incorporate *natural* base pairs with an optimized K_d , while no such optimization has occurred (of course) with *unnatural* base pairs. Indeed, we

repeatedly observed higher K_d value of other UBPs and RNAPs (for example, Ds:Pa on T7 RNAP, TPT3:NaM on RNAP II), suggesting a room for UBP design or RNAP engineering to further improving UBP incorporation efficiency.

To understand what could cause weaker binding, we first examined the active site of RNAP residues, in particular the residues near the minor groove (since **B** and **S** has distinct minor groove functional groups in comparison with natural nucleotides). In contrast to DNAPs-minor groove interaction (which occur via direct hydrogen bonding with polar or basic residues, such as Lys/Arg/Gln/Asn, with narrow minor groove in B-form DNA duplex), we found that the residues in multi-subunit RNAP at the minor groove edge of +1 template base and incoming substrate are mainly non-polar, hydrophobic residues (Pro 427, Ala 426, and Met 1273). These non-polar hydrophobic residues presumably make Van der Waals interactions (see new Extended Fig. 6). This finding leads us to believe that the differences in the interactions between the water solvent shell and the molecular systems (UBP/RNAP vs natural NTP/RNAP) may partially account for the K_d difference of UBP substrates.

1. The de-solvation energy of incoming UBP substrate or template (such as **BTP** or **dB**) at the minor groove edge might have a higher penalty than their natural substrate counterparts, i.e., these are better solvated when they are outside of the active site.
2. It is well-established that nucleic acid duplexes (including DNA duplexes, and RNA duplexes, as well as RNA:DNA hybrids) have a structure-specific minor groove hydration network with ordered water molecules, which makes important contributions to duplex stability¹⁻⁷. It is possible that the minor groove hydration pattern of nucleic acid duplex is different from the UBP-containing duplex.

Future studies are required to explore these possibilities.

Text should make clear that the PZ pair, AT gels, and B(imine tautomer) are present in Supporting information.

We have revised several sentences in the supporting information to ensure a clear presentation of the information.

Page 9 –provide information in the text about pH and temperature of the complex before freezing as these can affect tautomerization.

We added pH and temperature information of sample preparation in methods section.

- Fig 3. Should be Fig. 4 in the text (2 times)

We revised Fig. 3 to Fig. 4.

Page 15 – “cooled-down” instead of “cool-down”?

We revised to “cooled down” as suggested.

Page 16 – they should include the percentage of dPAGE gel

We now indicate that we used a 12% denaturing urea-PAGE gel, as suggested by reviewer.

- “Watson-Crick U’s enol form”, “wobble base pair U’s keto form”

We revised our manuscript as suggested.

Figure 1: Explain abbreviations in the legend – nts, ts

X: dB or dS (what about dT they checked with ATP?)

We added template strand and non-template strand for ts and nts in figure legend. Yes, we also performed transcription assay of ATP incorporation with dT template. We now added dT in Figure 1a as suggested by reviewer.

Figure 6: Add information to clarify what tautomerizes – “B tautomerization” “U tautomerization” instead of just “tautomerization”

We clarified B:U tautomer forms in Figure 6 and revised figure legend.

UTP is incorporated in a different manner than other nucleotides (several bands visible) – needs explanation.

The upper band (n+2) is due to the second UTP incorporation (opposite with downstream dA template at the n+2 position). We have included the following sentence to describe the result: “Interestingly, for UTP incorporation reaction, n+2 extension product (second UTP incorporation opposite the n+2 dA template) is also observed, suggesting efficient transcription extension from a B:U mispair.”

The table showing kinetic parameters should have an explanatory title/legend/ and footnotes.

We added Table 1 title and legend to figure legend part.

References

- 1 Conte, M. R., Conn, G. L., Brown, T. & Lane, A. N. Hydration of the RNA duplex r(CGCAAUUUGCG)₂ determined by NMR. *Nucleic Acids Res* **24**, 3693-3699, doi:10.1093/nar/24.19.3693 (1996).
- 2 Woods, K. K., Lan, T., McLaughlin, L. W. & Williams, L. D. The role of minor groove functional groups in DNA hydration. *Nucleic Acids Res* **31**, 1536-1540, doi:10.1093/nar/gkg240 (2003).
- 3 McDermott, M. L., Vanselous, H., Corcelli, S. A. & Petersen, P. B. DNA's Chiral Spine of Hydration. *ACS Cent Sci* **3**, 708-714, doi:10.1021/acscentsci.7b00100 (2017).
- 4 Geyer, C. R., Battersby, T. R. & Benner, S. A. Nucleobase pairing in expanded Watson-Crick-like genetic information systems. *Structure* **11**, 1485-1498, doi:10.1016/j.str.2003.11.008 (2003).
- 5 Chuprina, V. P. *et al.* Molecular dynamics simulation of the hydration shell of a B-DNA decamer reveals two main types of minor-groove hydration depending on groove width. *Proc Natl Acad Sci U S A* **88**, 593-597, doi:10.1073/pnas.88.2.593 (1991).
- 6 Gyi, J. I., Lane, A. N., Conn, G. L. & Brown, T. The orientation and dynamics of the C2'-OH and hydration of RNA and DNA.RNA hybrids. *Nucleic Acids Res* **26**, 3104-3110, doi:10.1093/nar/26.13.3104 (1998).
- 7 Chen, Y. Z. & Prohofsky, E. W. The role of a minor groove spine of hydration in stabilizing poly(dA).poly(dT) against fluctuational interbase H-bond disruption in the premelting temperature regime. *Nucleic Acids Res* **20**, 415-419, doi:10.1093/nar/20.3.415 (1992).

Reviewers' Comments:

Reviewer #1:

Remarks to the Author:

I appreciate the authors' responses to my previous comments. With the incorporated revisions, I support publication of the manuscript without further changes.

Reviewer #2:

Remarks to the Author:

I have no further comments - the authors have addressed all of the reviewer concerns...

Reviewer #3:

Remarks to the Author:

The authors have satisfactorily addressed my comments from the previous round of review.

NCOMMS-23-22857-A

REVIEWERS' COMMENTS

Reviewer #1 (Remarks to the Author):

I appreciate the authors' responses to my previous comments. With the incorporated revisions, I support publication of the manuscript without further changes.

Reviewer #2 (Remarks to the Author):

I have no further comments - the authors have addressed all of the reviewer concerns...

Reviewer #3 (Remarks to the Author):

The authors have satisfactorily addressed my comments from the previous round of review.

We express our gratitude to all three reviewers for their positive feedback.